# Multi-decadal initialized climate predictions using the EC-Earth3

# **global climate model**

- 3 Rashed Mahmood<sup>1,2</sup>, Markus G. Donat<sup>1,3</sup>, Roberto Bilbao<sup>1</sup>, Pablo Ortega<sup>1</sup>, Vladimir Lapin<sup>1</sup>, Etienne
- 4 Tourigny<sup>1</sup>, Francisco Doblas-Reyes<sup>1,3</sup>
- <sup>1</sup>Barcelona Supercomputing Center (BSC), Barcelona, Spain
- <sup>2</sup>National Center for Climate Research (NCKF), Danish Meteorological Institute, Copenhagen, Denmark
- <sup>3</sup>Catalan Institution for Research and Advanced Studies (ICREA), Barcelona, Spain
- Correspondence to: Rashed Mahmood (rama@dmi.dk)
- **Abstract.** Initialized climate predictions are routinely carried out at many global institutions that predict the climate up to
- next ten years. In this study we present 30 year long initialized climate predictions and hindcasts consisting of 10 ensemble
- members. We assess the skill of the predictions of surface air temperature on decadal and multidecadal timescales. For the
- 10 year average hindcasts, we find that there is limited added value from initialization beyond the first decade over a few
- regions, but no added value from initialization was found for the third decade (i.e. forecast years 21-29). The ensemble
- spread in the initialized predictions grows larger with the forecast time. However, the initialized predictions do not
- necessarily converge towards the uninitialized climate projections within a few years and even decades after initialization.
- There is in particular a long-term weakening of the Atlantic Meridional Overturning Circulation (AMOC) after initialization
- that does not recover within the 30 years of the simulations, remaining substantially lower compared to the AMOC in the
- uninitialized historical simulations. The lower AMOC mean conditions also result in different surface temperature anomalies
- and the second contract of the second contrac

over northern and southern high latitude regions with cooler temperature in the northern hemisphere and warmer in the

- 20 southern hemisphere in the later forecast years as compared to the first forecast year. The temperature differences are due to
- less transport of heat to the northern hemisphere in the later forecast years. These multi-decadal predictions therefore
- 22 highlight important issues with current prediction systems, resulting in long-term drift into climate states inconsistent with
- 23 the climate simulated by the historical simulations.

### 1 Introduction

- Global and regional climate is changing due to anthropogenic emissions of greenhouse gases, atmospheric aerosols and
- internal climate variability (Chung and Soden, 2017; Bonfils et al., 2020; Chiang et al., 2021; IPCC, 2023). The ongoing
- increase in global temperature is causing devastating effects on life, ecosystems and important infrastructures requiring
- appropriate adaptation strategies to minimize the potential future effects (Gampe et al., 2021; Naumann et al., 2021; Brullo

1

2

19

et al., 2024). Accurate and reliable information about the near-term (10 to 30 years) future climate could underpin such adaptation efforts, obtained either from historical and projection climate simulations (hereafter also referred to as "uninitialized projections", as these simulations are not started from observational climate states) or from initialized decadal climate predictions. The long-term uninitialized projection simulations experience various types of uncertainties including the modeled forced response, assumptions about expected future conditions, and internal climate variability (e.g. Lehner et al., 2020). In particular for the near-term regional climate projections, internal climate variability can be the largest source of uncertainty (Hawkins and Sutton, 2009; Lehner et al., 2020) since these model simulations are started from different climate states during the preindustrial time.

The initialized decadal climate predictions are aimed at aligning the phase and amplitude of variability between numerical simulations and observations (e.g. Meehl et al., 2021), and initialization has been shown to correct forced model responses, by starting the simulations from observationally-constrained climatic states (Smith et al., 2007; Keenlyside et al., 2008; Doblas-Reyes et al., 2013). For the most recent version of the Coupled Model Intercomparison Project phase 6 (CMIP6), such model simulations were typically initialized every year and forecasts were made up to 10 years after initialization (Boer et al., 2016). On multi-annual and decadal timescales, the initialized climate predictions have shown skill in predicting climate over global regions (Smith et al., 2019, 2020; Delgado-Torres et al., 2022). In particular for near-surface air temperature predictions, the initialized predictions have also shown added value from initialization over the uninitialized projection simulations on multi-annual timescales in some regions (e.g. Smith et al., 2019; Delgado-Torres et al., 2022).

Initialized climate predictions are computationally expensive to run and therefore most prediction systems only forecast up to 10 years after initialization (Boer et al., 2016). Some recent studies have suggested that there is enhanced skill beyond 10 years from internal climate variability. This has been found, for example, by combining information from decadal predictions and longer-term uninitialized projections (Befort et al. 2022) and constraining patterns of sea surface temperature anomalies in large ensembles of the uninitialized projection simulations (Mahmood et al., 2021, 2022; Donat et al., 2024). Mahmood et al. (2022) demonstrated added skill over forcing alone for 20-year constrained projections. Similarly, Donat et al. (2024) found enhanced climate predictability up to 20 years when constraining climate variability in large multi-model ensembles from CMIP6, basing the constraints on either observations or decadal predictions. While initialized predictions beyond 20 years were considered during the Coupled Model Intercomparison Project phase 5 (CMIP5), these simulations were restricted to only three start dates (Meehl and Tang, 2012), which did not allow for robust estimates of skill over such longer forecast horizons. Düsterhus and Brune (2023) recently produced initialized predictions out to 20 years with the MPI-ESM model, and concluded that the information from initialization may not necessarily be lost even up to 20 years after initialization, resulting in skillful predictions of surface air temperature beyond 10 forecast years over a few regions. Recent work by Deser et al. (2025) with idealised experiments further suggests that initialized internal variability can constrain

surface climate variability for multiple decades, highlighting the potential for extended predictability when key modes of variability are properly captured.

Motivated by the prospect of multi-decadal predictability from model initialization and with the aim to understand the model dependence in a previous study (Düsterhus and Brune, 2023), in this study we performed a set of initialized climate predictions with the EC-Earth3 model with a forecast horizon of 30 years. These predictions build on and extend decadal predictions produced with the same model (Bilbao et al. 2021). These experiments allow us to further study climate predictability on multi-decadal time scales, and more generally to assess initialized climate predictions and uninitialized projections as complementary sources of climate information for near-term climate change estimates beyond the next decade.

The primary goal of the current study is to introduce the initialized 30-year long predictions, which were performed with a state-of-the art forecast system and involve a large number of hindcasts, considering together the forecast range, number of initialization years and number of members per forecast. Additionally, we address several specific research questions related to the scientific relevance of the multi-decadal predictions, from whether the initialized predictions can be skillful in predicting surface temperature beyond 10 years after initialization, whether and when the spread in initialized predictions and the uninitialized projection simulations converge and if the drift in Atlantic Meridional Overturning Circulation (AMOC) continues after the first decade of forecasts which was discussed in Bilbao et al. (2021).

#### 2 Data and Methods

We performed a new set of multidecadal climate predictions with the Barcelona Supercomputing Center (BSC)'s decadal prediction system (Bilbao et al., 2021), which uses the CMIP6 version of the EC-Earth3 climate model (Döscher et al., 2022). This hindcast consists of a 10-member ensemble of 30-year predictions initialized on the first of November of every 5th year starting from 1960 to 2020 (i.e a total of 13 start dates predicting 30 years). Since the simulations start from November, the final forecast year (i.e. the 30<sup>th</sup> prediction year) finishes in October and therefore only 29 full years are available for analysis.

 This new hindcast uses the initial conditions from the current operational BSC decadal prediction system, which have been updated with respect to those from Bilbao et al. (2021). The new system uses interpolated initial conditions from the ERA5 atmospheric reanalysis (Hersbach et al., 2020; Soci et al., 2024), and ocean and sea ice initial conditions from a 5-member NEMO3.6-LIM3 simulation forced with historical ERA5 surface fluxes that assimilates ORA-S5 (Zuo et al., 2019) ocean temperature and salinity at the surface using a standard surface nudging approach with restoring coefficients of -200 Wm<sup>-2</sup>K<sup>-1</sup> and -750 mmd<sup>-1</sup>, respectively. Below the mixed layer, a Newtonian relaxation term is also applied to assimilate three-

dimensional EN4 (Good et al., 2013) temperature and salinity fields. For this, a relaxation timescale that increases monotonically with depth is used, which takes approximate values of 30 days at the subsurface, 700 at 1000m and 3300 at 5000m. The spread in the 10-member ensemble was generated by adding a small (i.e. on the order of  $10^{-5}$  K) perturbation to the 3-dimensional atmospheric air temperature, combined with the five ocean and sea ice initial conditions.

While decadal predictions using EC-Earth3 have been provided with annual initializations as a contribution to CMIP6 and to the WMO Lead Centre for Annual to Decadal Prediction (Hermanson et al., 2022), the new multi-decadal predictions presented in this study are initialized at reduced frequency (every fifth year) primarily for computational cost reasons. Given the high auto-correlation of the time series when predicting decadal to multi-decadal averages, this somewhat reduced initialization frequency does not affect the degrees of freedom for skill evaluation. Initializing only every five years, as opposed to every year, may however lead to missing potential windows of opportunity when the predictability is enhanced in relation to specific initial climate states (e.g. Liu et al. 2023) and to a certain level of aliasing (García-Serrano and Doblas-Reyes, 2012).

For reference and to evaluate added value (and possibly deteriorating effects) from initialization, we also use in this study a 10 member ensemble of CMIP6 historical and scenario simulations performed using the EC-Earth3 model. Both the transient climate simulations and initialized predictions use observed historical forcings up to the year 2014 and afterwards the estimates of forcings based on theSSP2-45 ("middle of the road") emissions scenario (O'Neill et al, 2016).

For skill evaluation, all model simulations were converted to a uniform five-degree grid. The skill of the predictions is evaluated based on anomaly correlation coefficient (ACC) and the residual correlations following Smith et al. (2019). The ACC assesses the common variability (both from internal and forced origin) between the observations and the simulations. The residual correlations indicate added value from initialization and are computed by correlating the residuals obtained after removing an estimate of the forced signal from predictions and observation using the ensemble mean of the historical simulations (Smith et al., 2019). We evaluate here the skill of the mean predictions for 10, 20 and 29 years after initialization. Specifically, we focus on averages of forecast years 1 to 10 (FY1-10), 11 to 20 (FY11-20), 21-29 (FY-21-29), 1 to 20 (FY1-20) and 11 to 29 (FY11-29). The observed surface temperature data used for evaluating the skill is from HadCRUT4.6 (Morice et al. 2012). The sea temperature observations used were obtained from the Met Office Hadley

Center's sea ice and sea surface temperature data (HadISST1.1; Rayner et al., 2003)

Evaluation of the initialized predictions is performed in anomaly space, with anomalies calculated based on forecast-time dependent climatologies to minimize the impacts related to the climate drift inherent in these systems (Meehl et al., 2014). We note that anomalies computed for initialized predictions with a reduced number of start dates (as it is the case in the current study) are subject to uncertainties given the small sample used for computing the climatologies. Here we compute

anomalies for each forecast month based on their respective climatologies from the seven start dates (every fifth start date from 1960 to 1990). The observed and historical simulation climatologies are computed for the same temporal period used to define the hindcast climatology. The statistical significance of the correlations and temperature differences is estimated by using a two tailed student's t-test.

Figure 1. ACC (left column) and residual correlation (right column) of the near-surface air temperature hindcasts for different forecast horizons. The first three rows show ACC and residual correlations for 10 year mean hindcasts while the last two rows are for 20 year mean hindcasts. Stippling indicate regions where ACC and residual correlations are not statistically significant at 95% confidence level.

### 3 Results

146147

## 3.1 Evaluating the skill of the initialized predictions

We first evaluate the skill of the initialized multi-decadal prediction system in predicting surface air temperature anomalies for different forecast periods (Fig. 1). The initialized predictions are generally skillful in simulating observed surface air temperature anomalies over most regions globally. For the first decade (i.e. FY1-10), the ACC over the Atlantic, Indian and parts of the Pacific ocean and many land areas is positive and statistically significant suggesting overall good correspondence with observations. These results are generally consistent with the ACC values computed from an updated version of the previous decadal hindcast system (Bilbao et al., 2021) that was initialized every year (Figure S1 a and S1b), in supplementary material) and using every 5<sup>th</sup> start-date (Figure S1c and S1d). When subsampling the system to use every 5<sup>th</sup> start-date (Figure S1c and S1d) the ACC loses significance in several regions in which also the new 30 year prediction system has no significant ACC. These results suggest that while overall patterns of skill remain similar in reduced initializations compared to annual initializations, however, there may be sampling uncertainties induced by the reduced number of initializations. For the second and the third decades (i.e. FY11-20 and FY21-29) the correlations between predictions and observations are generally higher than 0.7 over most regions of the globe. Over the eastern Pacific, however, the skill of the initialized predictions degrades after the first decade. We note that some of the differences in ACC between the three decadal forecasts are related to the fact that their evaluation periods are different (i.e. 1961-2000 for FY1-10, 1971-2010 for FY11-20 and 1981-2019 for FY21-29). Using the common analysis period (i.e. 1981-2020) for the three decadal mean predictions show ACC skill over similar regions in all three forecast periods with relatively more significant regions in the Atlantic ocean for the first decade (Fig. S2, in supplementary material). We note here that using the common analysis period for skill assessment also introduces uncertainties since different initializations are used for different forecasts. The multi-decadal forecast times (i.e. FY1-20 and FY11-29) show high correlations with the observations over the globe especially for FY1-20.

159160161

Since the ACC for temperature is strongly influenced by the model response to forcings such as e.g. the increase in GHG forcings, we use residual correlations following Smith et al. (2019) to evaluate the added value from initialization. We find here that the added value from initialization is largest in the first ten-year average hindcasts (i.e. FY1-10), but a few regions show added value for the second forecast decade (i.e. FY11-20). For the FY1-10 we find significant added value over the Atlantic ocean and parts of the Pacific and over land regions including northwest Canada, central USA, the Middle East and northern Australia. For the second decade, the added value from initialization is found only over a few regions in the Pacific ocean. The residual correlations are generally not statistically significant over global land areas for FY11-20. For the third forecast decade (i.e. FY21-29), almost no added value from initialization is found in any of the three main ocean basins, suggesting that the information from initialization is lost over time especially after the first decade of the forecasts. When using a common analysis period (i.e. 1981-2020), the added value even for the first decade (i.e. FY1-10) is also limited to

parts of the Atlantic ocean, Eastern USA and southern Indian ocean (Fig. S2) indicating sampling uncertainties due to using different forecast analysis periods. The multi-decadal forecast times show added value from initialization for FY1-20 while a widespread detrimental value is found for FY11-29. For FY1-20 most of the added value is in similar regions as the added value for the first decade although parts of tropical regions in the Pacific and Indian oceans seem to show significant added value only in 20-year mean predictions.

## 3.2 Time series analysis of the initialized predictions and the uninitialized projection simulations

To understand how the 30-year predictions compare with the corresponding (uninitialized) historical/scenario simulations, we evaluate here the time evolution of their regionally averaged sea surface temperature (SST). For this we focus on two regions of well-known multidecadal variability such as the Subpolar North Atlantic (SPNA: 45°N-60°N, 50°W-20°W) and the northeast Pacific (NEP: 40°N-55°N, 140°W-122°W). Previous studies have shown that the decadal predictions can be highly skillful in forecasting temperature over SPNA region (e.g. Yeager et al., 2018; Delgado et al., 2022). This is also a region strongly influenced by forecast drifts associated with the Atlantic Meridional Overturning Circulation (AMOC) in initialized predictions. initialized decadal prediction systems have also shown skill in predicting climate over NEP region (e.g. Kataoka et al., 2020; Choi and Son, 2022).

As expected from small differences in the initial conditions, the spread in the initialized predictions is small at the beginning and grows over the course of the simulations (Fig. 2). In the SPNA region, the SSTs tend to drift towards a lower (i.e. cooler) mean state a few years after initialization resulting in larger biases (Fig. 2). Interestingly, while the mean SSTs for both the initialized predictions and uninitialized projections are close to each other at the beginning of the predictions, however, the drift in initialized predictions leads to a different mean state after ~10-15 years which is completely outside the ensemble spread of the uninitialized projection simulations. We also note that the spread tends to be narrower at the end of the forecasts than for the uninitialized projection ensemble, in particular for the earlier start dates (i.e. 1960 to 1995). For the NEP region, the mean SSTs in the initialized predictions follow the observations closely at all forecast years while the uninitialized projection simulations tend to be biased high (i.e. warmer than observations and predictions) (Fig. 3). The ensemble spread in the predictions is generally similar to the spread in uninitialized projections for this region (Fig. 3). Compared to the SPNA region, the predictions do not show a strong drift after initialization for the NEP region.

The SST anomalies show that the initialized and uninitialized ensemble spreads overlap (Fig. S3 and S4). The corresponding plots also nicely illustrate that the ensemble spread for the SPNA region is generally smaller in the initialized predictions compared to the uninitialized projections while similar for the NEP region (Fig. S3 and S4). Due to the non-stationary drift in individual prediction simulations (i.e., the different start dates in Fig. 2), the temperature anomalies for the SPNA region tend to be higher than the projection ensemble in the most recent initializations (cf Figs. 2 and S3). For the NEP region,

which is not strongly affected by the drift, the temperature anomalies remain similar for all start dates (Fig. S 4). All these results suggest that the climate information from different experiments (i.e. initialized predictions and the uninitialized projections) of the same model may not converge even after several years of predictions, depending on the region.

# SPNA (tos): runavg:36mon

Figure 2. Comparison of mean SST time series in SPNA in initialized predictions (in pink) and the uninitialized projections (in blue). The colored polygons show spread in 10 ensemble members (i.e. based on minimum and maximum values of the 10 members). Observations are shown by the black lines. The monthly mean time series were smoothed using a 36 months running average. The yellow lines (right y-axis) represent spread ratios between initialized and uninitialized ensembles.

# NEP (tos): runavg:36mon

Figure 3. Same as Figure 2 but for the northeast Pacific (NEP) region.

### **3.3 AMOC**

The AMOC is one of the main drivers of decadal-to-centennial variability in the Atlantic Ocean and it is thought to be strongly predictable due to its slowly evolving modulations (e.g. Zhang et al., 2019). Previous studies have shown that, even though decadal prediction systems exhibit various AMOC errors associated with initialization shocks and model drift (Polkova et al., 2023), the effects of initialization on the AMOC can persist for longer than the length of the predictions, which are typically of 10 years (e.g., Bilbao et al., 2021).

We analyse the AMOC at 45°N, defined here as the overturning streamfunction value at 45°N and at 1000 m depth. The EC-Earth3 model produces a shutdown in Labrador Sea convection (and associated AMOC slowdown) also in some of the historical runs (Bilbao et al., 2021). We compare the predictions to the EC-Earth3 historical simulations run by BSC as part of CMIP6 simulations (a total of 15 members) for which we define two ensembles formed by members with (12 out of 15) and without (3 out of 15) deep ocean convection in the Labrador Sea region. These differences in historical simulations originate from the AMOC multi-centennial variability in the EC-Earth3 pre-industrial simulations (Meccia et al., 2023), from which the initial states are taken to initialize these runs.

Figure 4a shows that the 30-year predictions start from the reconstruction and then follow a similar evolution of the AMOC at 45°N (AMOC45) across all start-dates. This evolution is characterized by an initial strengthening in the first 2-3 years, followed by a rapid weakening, stabilizing at around 14 Sv within the first 25 forecast years, as shown by the climatological values (Fig. 4b). Comparing AMOC45 in the 30-year predictions with the current decadal prediction system shows that the climatology is well constrained, despite the different sampling of initial states. By contrast, the comparison with the previous forecast system shows differences in the AMOC45 during the first few years, since the reconstruction used in that system produces a stronger AMOC. Another difference is that the drift in the current system is rather consistent across start dates, which did not occur in the previous system (see Figure 6c in Bilbao et al., 2021), for which there was a different behavior pre and post-2000. Despite the updated initial conditions, the current system also suffers from the same initialization problem of the previous system (described in detail in Bilbao et al., 2021) where the model drift causes a progressive increase in Labrador Sea density stratification that ends up suppressing the mixed layer depth and weakening the AMOC. This effect persists beyond the first decade in these new predictions with updated initial conditions, and the AMOC does not recover

within the 30 forecast years to the historical mean state. We note that this systematic shutdown of convection in the forecasts is an artefact from initialization, as observational records in the Labrador Sea show instead intermittent deep-convection events, closely tied to interannual—to—decadal variability in the North Atlantic Oscillation, whose positive phase enhances winter mixed-layer deepening (Yashayaev & Loder, 2016).

Consistent with Bilbao et al. (2021), the predictions tend to drift towards a state closer to the historical simulations with suppressed convection in the Labrador Sea (purple), in contrast with those members that exhibit convection (green). However, the longer predictions also reveal that the decadal forecasts stabilize at an even weaker AMOC state (i.e. ~14 Sv) which is lower than the historical mean AMOC (and lower even than the mean of those historical runs with suppressed Labrador sea convection). Therefore, in contrast to previous expectations (i.e. based on 10 year predictions), we find here that the AMOC45 does not converge to the state in the historical simulations. This conclusion supports the previous findings of Düsterhus and Brune (2023) in their 20-year predictions in which the AMOC in initialized predictions tends to drift towards much lower values than the historical simulations albeit using a different climate model.

Figure 4. (a) Evolution of the AMOC at 45N in the 30 year prediction ensemble mean (blue to red every five start dates), the historical+projection ensembles with and without Labrador sea convection (green and purple, respectively) and the ocean reconstruction used to generate the initial conditions (black). (b) The climatological values as a function of forecast time. DP and DP\_30yrs in b represent decadal and the muliti-decadal (i.e. 30 year long) prediction systems respectively, both both of which use the same updated initial conditions. DP\_Old represents the previous decadal prediction system from Bilbao et al. (2021)

We find that surface temperature responses during the course of the simulations are strongly influenced by the prominent drift in the AMOC. The climatological mean surface air temperature differences between the later forecast years compared to the first forecast year (Fig. 5a-d) are negative in the northern hemisphere while positive in the southern hemisphere, a pattern

that is consistent with the reduced northward heat transport that follows an AMOC weakening. These bipolar temperature differences persist until decades after initialization and are consistent with the AMOC drift towards lower mean state. The temperature differences between forecast year 29 and 19 (Fig. 5e) are relatively small and symmetrical in both hemispheres, consistent with the stabilization of AMOC after the first two decades of initialization. The interhemispheric asymmetrical responses of surface air temperature due to AMOC decline has been found in previous studies (e.g. Orihuela-Pinto et al., 2022).

Figure 5. Differences in mean near-surface air temperature between different forecast years of the initialized predictions. Stippling indicates regions where the differences are not statistically significant at 95% confidence level.

#### 4 Summary and discussion

The initialized climate predictions are designed to synchronize the model variability with that of observations and thus can provide potentially more skillful climate information for the next few decades than the forced-only climate projections. Most of the current initialized climate prediction systems generally forecast up to ten years after initialization (e.g. Boer et al., 2016). Here we performed a new set of multidecadal climate predictions with the BSC's decadal prediction system using updated initial conditions. Ten-member ensembles of 30-year long hindcasts and predictions were performed using the EC-Earth3 model by initializing every fifth year starting from 1960 to 2020.

The evaluation of the multidecadal predictions suggest that the predictions are skillful for the different forecast periods considered (up to 30 years), however the added value from initialization was primarily found for the first 10 year average forecasts after which very limited added value from initialization is found. In particular for the third decade (i.e. FY21-29)

we did not find added value from initialization over the uninitialized projection simulations, and identify even some detrimental effects associated with long-term model drift after initialization. Similar issues pointing to long-term drift in AMOC were found in a previous study which presented 20-year initialized hindcasts with a different decadal prediction system based on the MPI-ESM climate model (Düsterhus and Brune, 2023). While the skill evaluations are subject to large uncertainties related to the limited sample size relative to the forecast times, an important result of this 30-year prediction experiment is the indication of long-term model drift into a different climate state characterised by weakened AMOC, that does not recover within the 30 years of the simulations.

As a consequence of this long-term drift the initialized predictions and the uninitialized projections may not necessarily provide consistent information for near-term climate change estimates. Also the ensemble spreads for the two types of simulations are non-stationary (especially for the SPNA region, Figure 2) and generally smaller in initialized predictions than the uninitialized projections. Besides the inconsistent near-term climate estimates from the initialized and uninitialized simulations, these results also imply that caution must be taken when combining climate information from these two types of model simulations. For example, these inconsistencies will challenge approaches to concatenate data from decadal predictions and climate projections (Befort et al., 2022).

Similar to the findings by Bilbao et al. (2021), we find that the climate predictions do not converge to the model attractor characterized by the historical ensemble, a deviation that is particularly evident for the AMOC. Indeed, the AMOC drifts in the first ~22 forecast years and then tends to stabilize at ~14 Sv which is lower than the mean AMOC state in non-initialized simulations. A similarly lower AMOC mean state in initialized predictions compared to historical simulations was also found by Düsterhus and Brune (2023) using a different climate model. This suggests that both models can experience different stable AMOC states, with initialization from observations playing a pivotal role to move from one state to the other.

The progression in the AMOC drift is found to have a strong influence on the predicted global surface air temperature anomaly pattern. The climatological mean temperatures in later forecast years compared to the first forecast year, tend to be cooler in the northern hemisphere and warmer in the southern hemisphere, as expected in response to an AMOC weakening, in line with results in idealized experiments enforcing an AMOC shutdown (Orihuela-Pinto et al. 2022).

One side effect of long prediction time horizons and using multiple ensemble members is the need for large amounts of computational resources for running simulations, post-processing data and the storage space requirement. Due to these reasons, the current prediction system was only initialized every fifth year which makes it challenging for computing anomalies and also evaluating the skill of the predictions. Alternative approaches for providing climate information on 10 to 20 year mean timescales have been developed which require a fraction of computational costs as compared to the state-of-the-art initialized climate predictions. These alternative approaches make use of existing climate simulations and constrain

them according to the similarity of their SST variability patterns with observations at a given initialization time. The climate predictions based on these variability-constrained projections (e.g. Mahmood et al., 2022; Donat et al., 2024) even show higher and more widespread added skill, compared to the initialized predictions presented in this study, for both 10 and 20 year mean predictions. The skill in these constrained projections suggests that the decadal and multi-decadal predictability of climate might be higher than what is achieved with the new initialised prediction using EC-Earth3. This warrants further indepth comparison of the different modelling and the constraining approaches in predicting climate over multidecadal timescales, ideally in a multi-model context.

While our results show very limited added predictive skill from initialization beyond the first decade, they highlight important issues often affecting initialized prediction systems. The model drift following an initialization shock can cause the model to adjust into a different climate state compared to uninitialized simulations for an extended time period. Our results show that even after 30 years there is no sign of convergence between the initialized and uninitialized climate simulations. This highlights important challenges in the use of initialized predictions for estimates of near-term climate changes, that should be taken into account in the further development of both the climate models and the initialization approaches.

### Acknowledgments

This research has been supported by the Horizon Europe project ASPECT (grant number 101081460). We are also grateful for support by the Departament de Recerca i Universitats de la Generalitat de Catalunya for the Climate Variability and Change (CVC) Research Group (Reference: 2021 SGR 00786). We are also grateful for partial support by the Horizon Europe project Impetus4Change (grant number 101081555). RM acknowledges partial funding from the National Center for Climate Research (NCKF) of the Danish Meteorological Institute (DMI). We acknowledge EuroHPC Joint Undertaking for awarding us access to MeluXina at LuxProvide, Luxembourg with the EuroHPC Regular Access computing project EHPC-REG-2023R01-125 "ASPECT-EC: Adaptation-oriented Seamless Predictions of European ClimaTe using the EC-Earth model"

- Data availability
- The data from these simulations are freely available at the ESGF archive.

### **Author contributions**

MD, FDR and RM designed the study. RM performed the climate model simulations. RM and RB analysed the resultss. RM, MD, RB and PO discussed the results for presentation in the study. RM and MD wrote the initial manuscript draft with contributions from all co-authors. VL, RB, and ET generated initial conditions and also developed capability to perform initialised predictions with the EC-Earth3 model.

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
