# Peer review of "Multi-decadal initialized climate predictions using the EC-Earth3"

_EGUsphere, 2025_

## Author Comment (AC1)

**Summary**

The study by Mahmood et al. makes an important step forward in understanding initialized prediction on decadal timescales (out to 30 years), a topic that is only rarely addressed due to computation costs. The results are very interesting, and ultimately seem to hinge on the fact that the initialized runs push AMOC into a weakened state that is outside the range of what is captured by the uninitialized simulations. The authors could enhance the impact of this study by focusing in a bit more on this topic – is this a realistic plausible reality? Why do these differences exist in the Lab Sea convection? And if initialized predictions are perhaps unreliable at this timescale (multi-decadal), should the community be pushing instead towards what the authors refer to as "variability-constrained projections" (which would benefit from even a brief description).

Author Response: We thank the reviewer for providing thoughtful comments on the paper which lead to improved manuscript. Following the reviewer's suggestions, we have expanded the discussion related to AMOC drift in initialized predictions. We have also expanded on the discussion related to variability constrained projections.

**Major Comments:**

- The discussion in the first paragraph of Section 3.1 should much more nuanced. The authors state that most land areas show ACC that is positive and statistically significant, with an exception of South America; But looking at Fig 1, there is *a lot* of stippling indicating things are not significant, even over land. Signals over Australia, for example, are rarely significant. In comparing with Fig S1, again it seems that significance has actually dropped dramatically in the case with fewer years (most notable over the Indian Ocean and Pacific). This calls into question the assertion that "a reduced sample size (due to initializing by every fifth year) does not strongly affect the skill of the prediction system."

Author Response: We agree with the reviewer that this section needed more attention which we have now modified. In particular we changed the text to reflect that the new prediction system has relatively fewer regions with significant ACC compared to the previous system. The modified text reads as: "These results are generally consistent with the ACC values computed from a updated version of the previous decadal hindcast system (Bilbao et al., 2021) that was initialized every year (Figure S1a and S1b), in supplementary material) and using every 5th start-date (Figure S1c and S1d). When subsampling the system to use every 5th start-date (Figure S1c and S1d) from the previous prediction system the ACC loses significance in several regions in which also the new 30 year prediction system has no significant ACC. These results suggest that while overall patterns of skill remain similar in reduced initializations compared to annual initializations, however, there may be sampling uncertainties induced by the reduced number of initializations."

- Figure 1: Given how much of the signal is not significant and thus stippled, I wonder if the fig would be more readable if stippling indicated where things *are* significant? It's hard to discern at first glance right now.

**Author Response:** The reason why we apply  stippling over the non-significant values is to not obscure the results which are significant, that are the ones we are really interested in. The symbols on top of the colored values make it harder to see shading below and identify the areas with maximum correlations, therefore we prefer to keep the stipplings on non-significant values.

- Discussion of Fig 2 notes the different mean state in initialized predictions, but should also discuss the much larger bias as a result.

**Author Response:** We modified the sentence discussing these results to indicate that the drift results in large biases in initialized predictions. The modified sentence reads as: " In the SPNA region, the SSTs tend to drift towards a lower (i.e. cooler) mean state a few years after initialization resulting in larger biases (Fig. 2)."

- The discussion around the Labrador Sea is interesting, as it points to inherent differences between initialized and uninitialized models. Some added context would strengthen the section further; for instance, is a shutdown in the Lab Sea convection rooted in any observations? Is it realistic, or an unwanted model behavior? *Why* does this happen in initialized runs?

**Author Response:** Observational records of the Labrador Sea show intermittent deep-convection events, closely tied to interannual–to–decadal variability in the North Atlantic Oscillation, whose positive phase enhances winter mixed-layer deepening (Yashayaev & Loder, 2016). By contrast, the convection shutdown seen in EC-Earth is connected to a fundamentally different mechanism, both in timescale and driving process. Meccia et al. (2023) investigates this mechanism at depth showing that it is driven by the accumulation of salinity anomalies in the Arctic and their release into the North Atlantic, which occur with a periodicity of ~150 years. Although such a centennial-scale process cannot be ruled out in the real ocean, instrumental records are too short to detect it and proxy data offer no clear evidence that it exists. Regarding why this shutdown occurs systematically in the initialized forecasts, Figure 11 of Bilbao et al. (2021) demonstrates that, upon initialization, the model rapidly transitions in surface density toward its historical climatology in the Labrador Sea, which occurs without a corresponding subsurface adjustment. This surface–subsurface imbalance amplifies stratification, prematurely halting deep convection, and trapping the model in a prolonged suppressed state. We therefore believe that the shutdown in the model is an unrealistic artefact.

This distinction between the observed NAO-driven intermittency versus model-specific, centennial-scale shutdown in initialized runs is now discussed in the revised manuscript (see lines 258-262).

- Line 305: "The climate predictions based on these variability-constrained projections (e.g. Mahmood et al., 2022; Donat et al., 2024) even show higher and more widespread added skill, compared to the initialized predictions presented in this study, for both 10 and 20 year mean predictions." – this seems like a huge point, but isn't backed up by comparison within this text. It would make a stronger paper overall to focus in on this as a piece of the larger manuscript.

**Author Response:** We have added a couple more sentences to expand the discussion related to this point. A comparison of results between variability constrained and initialized predictions is not included in this manuscript as it is out of the scope of the current study. However, a separate ongoing work using multi-decadal predictions from different climate models could provide a comparison between constrained projections and the initialization climate predictions.

**Minor Comments**

- Line 51: What is meant by "constraining patterns of climate variability in large ensembles of the uninitialized projection simulations"? Some added description would help.

**Author Response:** We modified the sentence to reflect that we use sea surface temperature anomalies. The modified sentence read as "This has been found, for example, by combining information from decadal predictions and longer-term uninitialized projections (Befort et al. 2022) and constraining patterns of sea surface temperature anomalies in large ensembles of the uninitialized projection simulations (Mahmood et al., 2021, 2022; Donat et al., 2024).".

- Line 79: "…initialized on the first of November every 5th year…" – is there any rationale for using November? Is this consistent with other experiments? Would differences be expected if initializing in spring/summer instead of fall/winter? (Perhaps the Lab Sea response would differ in particular?)

**Author Response:** Decadal predictions are typically initialized in the first of November to be able to predict the first winter completely, which is relevant for ENSO and NAO studies. Indeed, the new DCPP protocol for CMIP7 will strongly recommend that all systems are initialized on the 1st of November, to ensure consistency across models. In CMIP6 several decadal prediction systems were initialized in the 1st of January, which has hindered the studies focusing on NAO predictability to systematically exclude the first winter, that is the most predictable one. The DCPP protocol for

CMIP7 also includes a new experiment that focuses on multi-annual timescales (with lead times up to 28 months) and that will explore the dependence of the results to the month of initialization.

- Line 90: "…on the order of 10-5 K)" – to clarify, is this meant to be 1e-5? That seems like it might actually be rather large, but I'm typically working in uninitialized space, wherein 1e-14 perturbations are common. Could the authors comment on if this is a standard magnitude for initialized predictions?

**Author Response:** Yes, it was meant to be 1e-5K. This value is much smaller than the observational uncertainty and therefore, we do not expect large differences if an even smaller number is used.

- Line 106: "…to a uniform five-degree grid" – that's really coarse; is there a reason to convert to five degrees? Is this the native resolution of some runs vs others?

**Author Response:** This was done to compare with observational data from HadCRUT which comes with a 5 degree resolution. Besides this is the recommended practice for evaluating initialized climate predictions (e.g. Goddard et al., 2013) in order to minimize effects from small-scale noise in the identification of large-scale predictable signals.

- Line 141: "…evaluation periods are different (i.e. 1961-2020 for FY1-10, 1971-2020 for FY11-20 and 1971-2020 for FY21-29)." – Could you limit the evaluation period for FY1-10 to 1971 onwards arbitrarily, just to test the impact?

**Author Response:** We have now included a new supplementary figure (Figure S2) and also added related text in the revised manuscript. However, we also note that using a common analysis period introduces its own uncertainties since different initializations are needed for different forecast periods. See lines (157-162) and lines (173-175).

- Line 149: "…over land regions including northwest Canada and USA." – I'm not sure I see a lot of added value over the USA, it's mostly stippled? But Australia and the Middle East seem to show something more cohesive.

**Author Response:** There are several grid points which show significant residual correlation over Canada and central USA. We modified the sentence to indicate also added value over northern Australia and the Middle East. The sentence reads as: "For the FY1-10 we find significant added value over the Atlantic ocean and parts of the Pacific and over land regions including northwest Canada, central USA, the Middle East and northern Australia."

- Line 162: "Previous studies have shown that the decadal predictions can be highly skillful in forecasting temperature over SPNA region." – Citations?

**Author Response:** Citations added in the revised version.

- Fig 2: Please add a legend for the lines, in addition to the description the caption for easier readability.

**Author Response:** Agree, legend is added in new figures.

- Fig 5: Suggest adding stippling for significance, as in other maps

**Author Response:** Agree. Stippling is now added over the regions where differences are not statistically significant.

---

## Author Comment (AC2)

**Referee 2:**

The added value of initialization for long (multidecadal) climate outlooks remains unclear, so this analysis of 30-year initialized predictions using the EC-Earth3 model will be of considerable interest to the decadal prediction community. The manuscript is clear and concise, and the results are intriguing. I recommend this be published after some minor issues are addressed. Most importantly, I think an additional analysis is needed (perhaps in supplemental) that shows how skill varies with lead time when a common verification window is used.

**Author Response:** We thank the reviewer for taking time and providing very constructive feedback on our study. We have revised the manuscript by taking into account all the comments and suggestions provided by the referee. Below we provide a point-by-point response to each of the referee comments.

Minor Comments

L48-60: A recent study suggests that initialized internal variability could constrain surface climate variability for multiple decades (Deser et al., 2025, https://doi.org/10.1007/s00382-024-07553-z). It may be worth including such potential predictability examples in this intro section.

**Author Response:** Added a new sentence citing Deser et al. (2025) as: "Recent work by Deser et al. (2025) with idealised experiments further suggests that the initialized internal variability can constrain surface climate variability for multiple decades, highlighting the potential for extended predictability when key modes of variability are properly captured."

L62: Unclear what "model dependence in previous results" refers to exactly

**Author Response:** We agree that it was not fully clear. It was meant for model dependence compared to the model used by Düsterhus and Brune (2023). We modified the sentence to indicate this as: "Motivated by the prospect of multi-decadal predictability from model initialization and with the aim to understand the model dependence in previous study (Düsterhus and Brune, 2023), in this study we performed a set of initialized climate predictions with the EC-Earth3 model with a forecast horizon of 30 years".

L112: If hindcasts are 30 years long, why not examine FY21-30 and FY11-30, for consistency?

**Author Response:** The predictions are initialized from November and run for 30 years, however the final prediction year runs until October. In order to analyze annual means (from January to December), we have one less year for the analysis. We agree with the reviewer that this information was not written explicitly in the manuscript. We have now clarified in the revised version of the manuscript about the availability of 29 full prediction years for analysis purposes.

L121: Should mention that observed anomalies are computed in similar fashion (observed climatology matching months/years used for hindcast climatology).

**Author Response:** Agree. We added a new sentence indicating this as: "The observed and historical simulation climatologies are computed for the same temporal period used to define the hindcast climatology."

Fig. 1: Caption should state the field being analyzed. Methods should clarify how statistical significance is quantified.

**Author Response:** Agree. Added near-surface air temperature in the figure caption. Also added information on how the statistical significance is computed in the methods section.

L137: Authors have an opportunity here to directly address skill sensitivities to sampling and system design choices, so why not do that in Fig. S1? A first FY1-10 ACC plot could subsample the previous system, selecting only start dates used in Fig. 1 (this would isolate the effects of changes in initialization, I believe). A second FY1-10 ACC plot could show skill when all start dates are included from the previous system (which would reveal the effect of reduced hindcast sampling on skill).

**Author Response:** We have now included in supplementary materials figures using all startdates and every 5th start-date from the 10 year prediction system. We have also added some discussion related to this at lines 145-150 in the revised manuscript.

L142: I think this should read "1981-2020 for FY21-29"? As noted above, the use of 9-year average (instead of 10-year average) for FY21-29 introduces another avoidable difference. The differences in verification window are likely important, so why not include a supplemental figure that shows skill for a common verification window (1981-2020 for all leads)? Otherwise, the conclusions that can be drawn from this comparison are very limited.

**Author Response:** Thank you. The description of years is corrected now. We have now included a new supplementary figure (Figure S2) and also added related text in the revised manuscript.

However, we also note that using a common analysis period introduces its own uncertainties since different initializations are needed for different forecast periods. See lines (157-162) and lines (173-175).

L152: I think the conclusions about impact of initialization are somewhat shaky given the discrepancies in verification window (see above).

**Author Response:**

L187: I would delete "as it was assumed in the past", because I don't consider fast convergence to be a generally held assumption, particularly for the North Atlantic.

**Author Response:** Agree. The text is deleted.

L213-215: Repetitive with introduction.

**Author Response:** Agree, the text is now removed from the revised version of the manuscript.

Fig. 4: Is it surprising that the AMOC climatology in DP_30yrs is almost identical to that from the DP (10-year), even at short leads, despite the different initialization methods?

**Author Response:** Both DP_30yrs and DP are initialized exactly in the same way. To clarify this we have added the AMOC climatology of the previous DP system. Comparing AMOC45 in DP_30yrs and DP shows that the climatology is well constrained even with fewer start dates. Comparing with the previous system shows that the updated initial conditions have an impact initially but overall the current system suffers from the same problems and the AMOC climatology evolution is the same. We have added a short paragraph in the text on this.

---

## Author Response (AR2)

**Dear Authors,**

Thank you for having provided a careful set of revisions. Prior to accepting your manuscript for publication I still have two queries. The first is very minor and concerns the reply to Reviewer #1's comment on Line 106. It may be worth adding a sentence reflecting your reply in the text, as several readers may indeed wonder about this choice. The second concerns the statistical significance computation. I appreciate that you have now added a brief description of this, but I note that you do not mention whether/how you have taken into account the issue of multiple testing. In your geographical maps, you are conducting a very large number of statistical tests on spatially correlated data, which may lead to an overstatement of significance if no correction to the significance level is applied (see e.g. Wilks, 2016:

https://journals.ametsoc.org/view/journals/bams/97/12/bams-d-15-00267.1.xml).

**AuthorResponse**: We thank the editro for providing additional feedback on our paper. We have now taken into account both points and revised the manuscript accordingly. For the first point: we have now included in the text justification related to the choice of using five-degree grid for regriding. The revised text reads as (lines 113:115):

"For skill evaluation, all model simulations were converted to a uniform five-degree grid in order to minimize effects from small-scale noise in the identification of large-scale predictable signals, recommended practice for evaluating the initialized climate predictions (e.g. Goddard et al., 2013)"

For the second point we have applied false detection rate to inorder to test for the multiple testing following recommendations from Wilks, 2016. Please see lines (133-134) which read as: "The statistical significance of the correlations and temperature differences is estimated by using a two tailed student's t-test. We apply the false detection rate (FDR) proceedure to test for multiple testing (Wilks, 2016) using alphaCDR=0.1."